# Representativeness in population-based studies of older adults: five waves of cross-sectional examinations in the Gothenburg H70 Birth Cohort Study

Hanna Wetterberg [1,2] Lina Rydén [1,2] Felicia Ahlner [1,2] Hanna Falk Erhag [1,2] Pia Gudmundsson [1,2] Xinxin Guo [1,2,3] Erik Joas [1,2] Lena Johansson [1,2] Silke Kern [1,2,4] Madeleine Mellqvist Fässberg [1,2] Jenna Najar [1,2,4] Mats Ribbe [1,2] Therese Rydberg Sterner [1,2,5] Jessica Samuelsson [1,2] Simona Sacuiu [1,2,4] Robert Sigström [1,2,4] Johan Skoog [1,2] Margda Waern [1,2,6] Anna Zettergren [1,2] Ingmar Skoog [1,2,4]

HW and LR are joint first authors.

For numbered affiliations see end of article.

**Correspondence to**
Hanna Wetterberg;
hanna.wetterberg@gu.se

## ABSTRACT

**Objectives** To describe representativeness in the Gothenburg H70 1930 Birth Cohort Study.

**Design** Repeated cross-sectional examinations of a population-based study.

**Setting** Gothenburg, Sweden.

**Participants** All residents of Gothenburg, Sweden, born on specific birth dates in 1930 were invited to a comprehensive health examination at ages 70, 75, 79, 85 and 88. The number of participants at each examination was 524 at age 70, 767 at age 75, 580 at age 79, 416 at age 85, and 258 at age 88.

**Primary outcome measures** We compared register data on sociodemographic characteristics and hospital discharge diagnoses between participants and (1) refusals, (2) all same-aged individuals in Gothenburg and (3) all same-aged individuals in Sweden. We also compared mortality rates between participants and refusals.

**Results** Refusal rate increased with age. At two or more examination waves, participants compared with refusals had higher educational level, more often had osteoarthritis, had lower mortality rates, had lower prevalence of neuropsychiatric, alcohol-related and cardiovascular disorders, and were more often married. At two examination waves, participants compared with same-aged individuals in Gothenburg had higher education and were more often born in Sweden. At two examination waves or more, participants compared with same-aged individuals in Sweden had higher education, had higher average income, less often had ischaemic heart disease, were less often born in Sweden and were more often divorced.

**Conclusions** Participants were more similar to the target population in Gothenburg than to refusals and same-aged individuals in Sweden. Our study shows the importance of having different comparison groups when assessing representativeness of population studies, which is important in evaluating generalisability of results. The study also contributes unique and up-to-date knowledge about participation bias in these high age groups.

### STRENGTHS AND LIMITATIONS OF THIS STUDY

⇒ The use of multiple comparison groups provides useful information on how to quantify the representativeness of population-based studies on older adults.

⇒ Included in the comparisons are a broad range of register-based variables, providing information on the differences and similarities between the groups.

⇒ The data were received on an aggregated level due to ethical regulations and hence only univariate analyses were possible to perform.

## INTRODUCTION

Cross-sectional, population-based studies are important to measure the societal burden of physical and mental disorders and the associated factors, as well as to provide baseline data for incidence studies.[1 2] Biased participation decreases representativeness and may affect the results of studies and lead to spurious findings. This occurs when those more prone to accept have specific characteristics.[2] One well-known potential bias is the 'healthy volunteer effect', where participants have a lower disease burden than refusals. Another factor known to decrease representativeness is women and individuals with higher socioeconomic status and higher education tending to participate more often.[1 3] Studies with low response rates are more vulnerable to self-selection bias than studies with high response rates.

Representativeness could be examined in different ways. One is to compare the characteristics between participants and refusals. Another is to compare participants with the target population from which the sample

was drawn. The third is to compare participants with the population of a larger geographical area, such as the country where the study was performed.

The Gothenburg H70 Birth Cohort Studies (the H70 studies) are multidisciplinary population studies examining birth cohorts of older populations in Gothenburg, Sweden. The aim is to study ageing and the prevalence of mental and physical disorders and their risk factors, considering the complex interactions with age, sex, gender, socioeconomic gradients, secular changes, as well as psychosocial, neurobiological and genetic factors occurring across the life course.[4] The studies started in 1971,[5] and since then five birth cohorts of adults aged 70 years old[4] and three birth cohorts of adults aged 85 years old[6] have been examined and followed longitudinally. The present study is concerned with the 1930 cohort, where we previously have published more than 30 papers based on cross-sectional examinations. However, no study has examined the representativeness of this cohort in detail, or made comparisons with the whole populations of Gothenburg and Sweden, or given a detailed overview of sampling methods for all examinations performed at ages 70, 75, 79, 85 and 88. This type of information is important to evaluate the external validity of a study and is essential to assess if the representativeness of studies changes over time.[7] This setting gives a unique opportunity to evaluate in detail participation bias in an older population.

This paper aimed to describe the representativeness of participants of the examinations of the birth cohort born in 1930, at ages 70, 75, 79, 85 and 88 years. We studied three levels of representativeness: in relation to refusals, same-aged individuals in Gothenburg and same-aged individuals in Sweden.

## METHODS

The H70 1930 birth cohort was examined at five occasions. All samples included residents of Gothenburg born in 1930, who were systematically selected based on birth dates. An exception was in 2009–2011, when only those invited previously were included. Individuals who were not traceable or did not speak Swedish were excluded from the eligible samples. The invitation procedure has been nearly identical at all examination years. An invitation letter was sent to all sampled individuals, including information about the study and a consent form. After 1–2 weeks, the research staff contacted the invited individuals by telephone. In cases of no contact or when the telephone number could not be found, reminders were posted. Individuals who were not able to visit the outpatient clinic were offered a home visit.

The H70 1930 birth cohort includes participants born in 1930 from the Prospective Population Study of Women (PPSW), a longitudinal study with baseline examination in 1968–1969.[8] The number of participants examined before 2000 within the PPSW study was 173 (33%) in

2000, 155 (20%) in 2005, 116 (20%) in 2009, 81 (20%) in 2015 and 53 (21%) in 2018.

## Data collection procedures

### Examinations

The examination procedures have been similar at all examinations[4] and include a 6-hour to 8-hour general examination. In addition, participants were invited to a range of additional examinations and a close informant interview. See online supplemental table S1 for details regarding the specific examinations conducted at the different waves.

### Comparison groups

We investigated the representativeness of the participants in the study in comparison with the refusals, same-aged individuals in Gothenburg and same-aged individuals in Sweden. Included in the comparison groups of Gothenburg and Sweden were all individuals born in 1930 who were registered residents of the municipality of Gothenburg and of Sweden, respectively. Hence, in the Gothenburg group, individuals who participated in the study were included as well. These data were received from Statistics Sweden on an aggregated level.

### Outcome variables

Information on the sociodemographic variables of participants, refusals and same-aged individuals from the municipality of Gothenburg as well as Sweden was received on an aggregated level from Statistics Sweden, and included marital status (married/widowed/divorced/never married), highest attained educational level (elementary/upper secondary/higher), country of birth (Sweden, yes/no), paid labour (occupational status, yes/no) and average individual income per month (mean and SD).

Hospital discharge diagnoses were obtained from the inpatient part of the National Patient Register (NPR), using codes from the Swedish version of the International Classification of Diseases (ICD)-9 and ICD-10. Diagnoses from 1987 up to each examination year were used in the analyses. Data were available at an individual level for participants and refusals in 2000, 2005 and 2009. Data on same-aged individuals living in Gothenburg and Sweden in 2000, 2005, 2009, 2015 and 2018 and data on participants and nonparticipants at the examinations in 2015 and 2018 were received on an aggregated level from the National Board of Health and Welfare. The NPR gained full national coverage in 1987 and all citizens in Sweden have access to medical care and therefore equal chance to be included in the NPR.[9] The hospital discharge diagnoses included cancer, neuropsychiatric disorders (including dementia disorders), cardiovascular diseases, diabetes mellitus, unipolar depression, alcohol-related disorders, ischaemic heart disease, cerebrovascular diseases, chronic obstructive pulmonary disease and osteoarthritis (see online supplemental table S2 for ICD codes).

Dates of death were available from the Swedish Tax Agency.

## Statistical analyses

Differences in sociodemographic factors and hospital discharge diagnoses between participants and refusals, same-aged individuals in Gothenburg and same-aged individuals in Sweden were compared using Pearson's $\chi^2$ test or Fisher's exact test for categorical variables and independent samples t-test for continuous variables. As the data were received on an aggregated level, only univariate analyses were performed.

Cox proportional hazards models adjusted for sex were used to compare mortality between participants and refusals, presented as HR and 95% CI. The risk time was calculated as the surviving time up to 5 years from age 70, 75, 79 and 85, and 2 years from age 88. Individuals who emigrated during follow-up were excluded from mortality analyses as their status is not known (four from examination in 2000, two from 2005, none from 2009, and one each from 2015 and 2018). The proportional hazard assumption was verified using Schoenfeld residuals. Mortality analyses were performed in R (V.3.5.3) using survminer and survival packages. P<0.05 (two-tailed) was considered statistically significant in all analyses.

This report complied with the Strengthening the Reporting of Observational Studies in Epidemiology guidelines for observational studies (including cross-sectional studies), as stated in the research checklist.[10]

## Participant and public involvement statement

There was no participant and public involvement in the study.

## RESULTS

The sample flow chart is displayed in figure 1.

In 2000, 775 residents of Gothenburg aged 70 years old were sampled. Among these, 753 were eligible (see figure 1) and 524 accepted to participate (response rate: 69.6%). Of these, 14 (2.7%) were examined at their home. The study was conducted from 4 September 2000 to 8 March 2002.

In 2005, 1250 residents of Gothenburg aged 75 years old were sampled. Among these, 1196 were eligible (see figure 1) and 767 accepted to participate (response rate: 64.1%). Of these, 117 (15.3%) were examined at their home. The study was conducted from 6 September 2005 to 20 March 2007.

In 2009, 1056 residents of Gothenburg aged 79 years old who had been invited at the 2000 or 2005 examination were invited. Among these, 950 were eligible (see figure 1) and 580 accepted to participate (response rate: 61.1%). Of these, 144 (24.8%) were examined at their home, while 6 (1.0%) had a combination of clinic and home examinations. The study was conducted from 6 November 2009 to 19 September 2011.

In 2015, 764 residents of Gothenburg aged 85 years old were sampled. Among these, 672 were eligible (see figure 1) and 416 accepted to participate (eligible response rate: 61.9%). Of these, 174 (41.8%) were examined at their home, while 18 (4.3%) had a combination of clinic and home examinations. The study was conducted from 7 October 2015 to 29 May 2017.

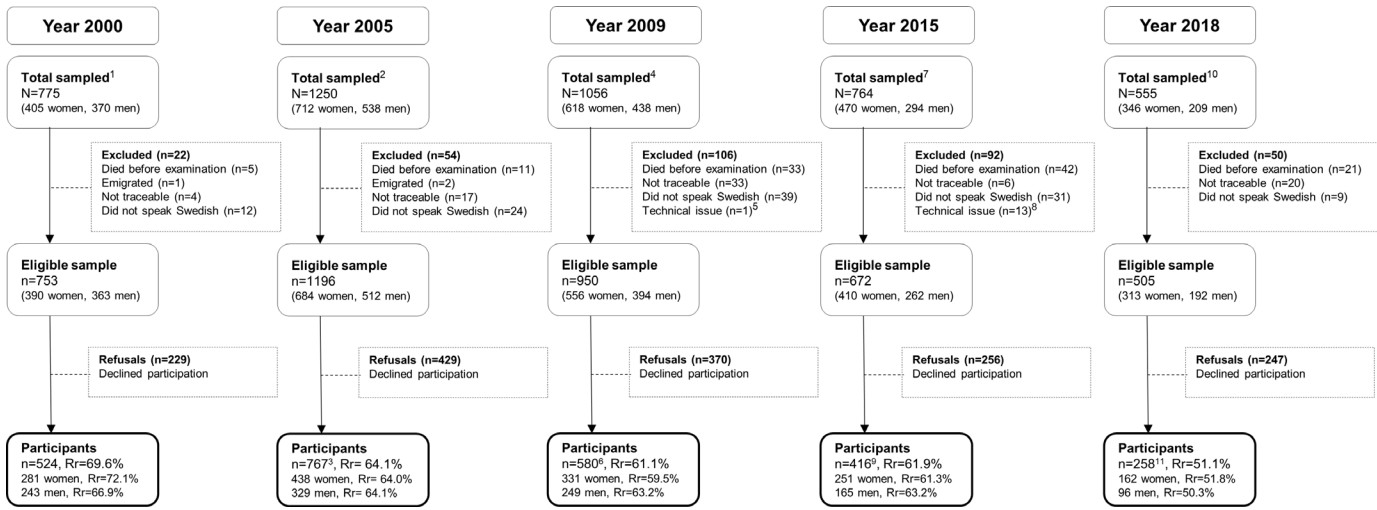

**Figure 1** Sample flow chart describing the five cross-sectional examination waves. [1]All 70-year-olds living in Gothenburg and born in 1930 on days 3, 6, 12, 18, 21, 24 or 30. [2]All 75-year-olds living in Gothenburg and born in 1930 on days 2, 3, 5, 6, 11, 12, 16, 18, 20, 21, 24, 27 (in January–May) or 30. [3]n=386 had previously been examined at age 70. [4]Included only individuals who had been invited to the 2000 or 2005 examination. [5]Misclassified as not living in Gothenburg and therefore not invited. [6]n=539 had previously been examined (n=307 at age 70, n=511 at age 75 and n=279 in both). [7]All 85-year-olds living in Gothenburg and born in 1930 on days 2, 3, 5, 6, 11, 12, 16, 18, 20, 21, 24, 27 or 30. [8]Misclassified as not living in Gothenburg and therefore not invited. [9]n=370 had previously been examined (n=199 at age 70, n=338 at age 75, n=337 at age 79 and n=174 in all three). [10]All 88-year-olds living in Gothenburg and born in 1930 on days 2, 3, 5, 6, 11, 12, 16, 18, 20, 21, 24, 27 or 30. [11]n=247 had previously been examined (n=120 at age 70, n=205 at age 75, n=207 at age 79, n=224 at age 85 and n=102 in all four). Rr, response rate (participants/eligible sample).

**Table 1** Comparison of response rates between examinations

| Examination year (response rate, %) | P value (2000) | P value (2005) | P value (2009) | P value (2015) | P value (2018) |
|---|---|---|---|---|---|
| 2000 (69.5) | – | 0.013 | <0.000 | 0.003 | <0.000 |
| 2005 (64.1) | | – | 0.151 | 0.381 | <0.000 |
| 2009 (61.1) | | | – | 0.690 | <0.000 |
| 2015 (62.0) | | | | – | <0.000 |
| 2018 (51.2) | | | | | – |

Statistical analysis: to test differences between groups, $\chi^2$ test was used.

In 2018, 555 residents of Gothenburg aged 88 years old were sampled. Among these, 505 were eligible (see figure 1) and 258 accepted to participate (eligible response rate: 51.1%). Of these, 131 (50.8%) were examined at their home, while 11 (4.3%) had a combination of clinic and home examinations. The study was conducted from 24 September 2018 to 20 December 2019.

The mean age of those examined was 70.6 years (SD 0.3) in 2000, 75.7 years (SD 0.4) in 2005, 80.1 years (SD 0.2) in 2009, 86.0 years (SD 0.2) in 2015 and 88.6 years (SD 0.2) in 2018. Among those who participated, 74.1% (n=811) participated more than once.

The response rate was higher in 2000 and lower in 2018 compared with all other examinations (table 1). There were no significant differences in response rates between women and men, although women had slightly higher response rates in 2000 (72.1% vs 66.9%, p=0.122) and lower in 2009 (59.5% vs 63.2%, p=0.254) (see online supplemental figure S1). For response rates in various subgroups, see online supplemental figure S1. The proportion excluded from the sample because they did not speak Swedish, died after sampling or were non-traceable was higher in 2009, 2015 and 2018 compared with 2000 and 2005 (online supplemental table S3).

### Mortality
Mortality was higher in refusals compared with participants at all examination years (figure 2). The test of Schoenfeld residuals indicated deviation from the proportional hazards assumption in the survival analysis in 2005, showing that the higher mortality among refusals was limited to the first 2 years (figure 2).

### Sociodemographic factors and hospital discharge diagnoses from registers
Table 2 shows the sociodemographic factors and hospital discharge diagnoses for participants, refusals and all same-aged individuals living in Gothenburg and in Sweden during the corresponding year. Analyses stratified by sex are shown in online supplemental tables S4–S8.

### Participants compared with refusals
Compared with refusals, participants had higher educational level and more often had osteoarthritis at all examinations, less often had neuropsychiatric disorders in 2000, 2005 and 2018, less often had cardiovascular disorders in 2000 and 2005, were more often married in 2000 and 2005, had higher average income per month in 2009, less often had alcohol-related disorders in 2000 and 2009, were more often born in Sweden, less often had cerebrovascular diseases, chronic obstructive pulmonary disease and dementia in 2000, and less often had unipolar depression in 2005 (table 2).

### Participants compared with the target population of same-aged individuals in Gothenburg
The participants comprised 14.9% (2000), 24.9% (2005), 21.9% (2009), 23.0% (2015) and 19.5% (2018) of the target population, that is, all same-aged individuals in the municipality of Gothenburg.

Compared with same-aged individuals in Gothenburg, participants had higher educational level in 2009 and 2015, had higher average income per month in 2000, were more often born in Sweden in 2009 and 2015, were more often widowed in 2018, and less often had cerebrovascular and neuropsychiatric disorders in 2000 (table 2).

### Participants compared with same-aged individuals in Sweden
Compared with same-aged individuals in Sweden, participants had higher educational level at all examinations, were less often born in Sweden at all examinations except in 2015, had higher average income per month in 2000 and 2009, were more often divorced in 2000 and 2005, less often had ischaemic heart disease in 2015 and 2018, more often had cancer in 2005, less often had cardiovascular diseases in 2018, and less often had cerebrovascular diseases in 2000 (table 2).

### DISCUSSION
We examined a birth cohort from Gothenburg, Sweden, at repeated cross-sectional examinations from ages 70 to 88 and found that the response rate declined with age. Participants were examined regarding three levels of representativeness: in relation to refusals, in relation to same-aged individuals in Gothenburg (the target population) and in relation to same-aged individuals in Sweden. We found a number of differences in sociodemographic factors and diseases in registry data between participants and refusals, as well as a lower mortality rate among participants. While similar proportions were observed in participants and the target population in Gothenburg for several factors, proportions differed when comparing

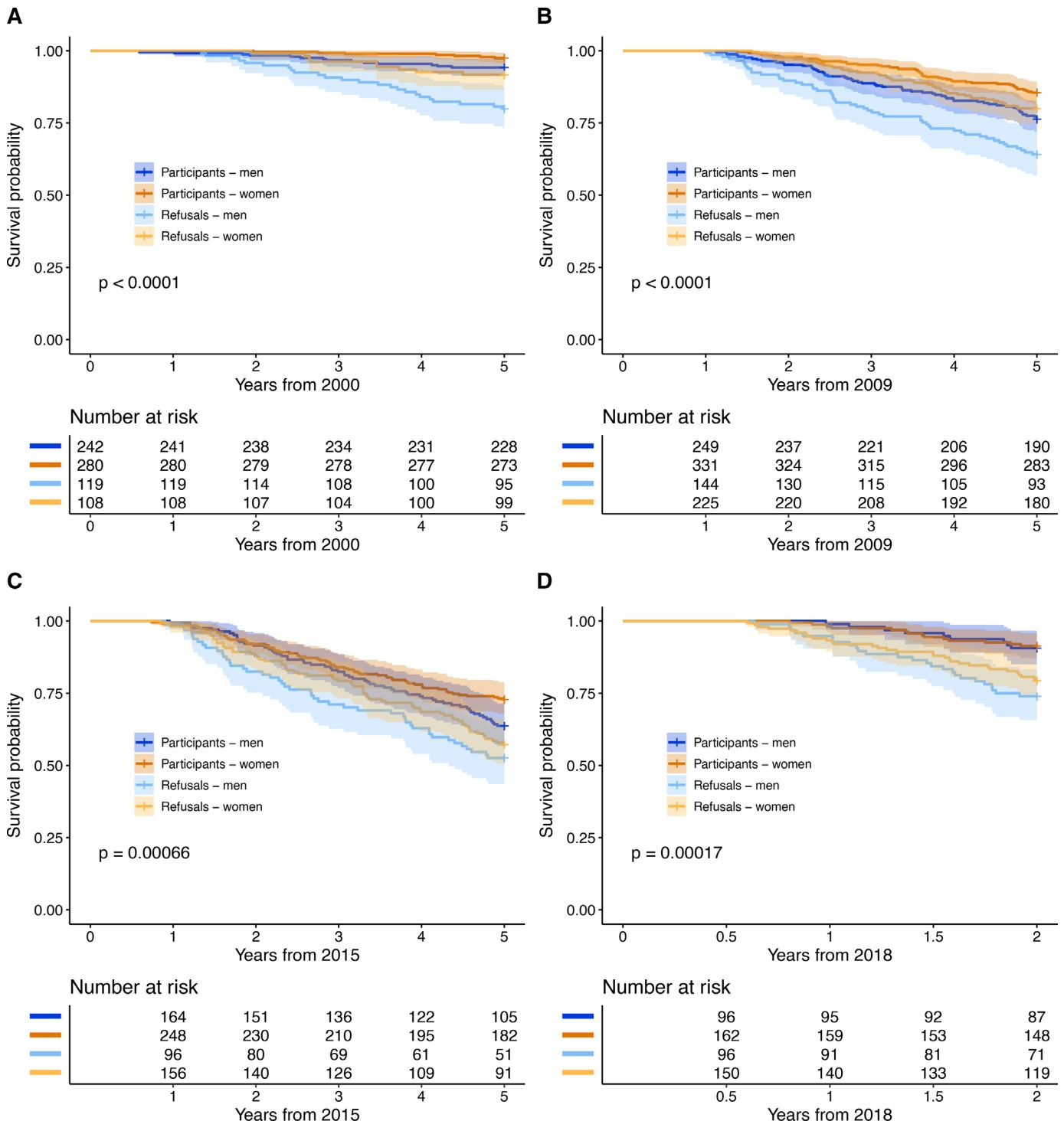

**Figure 2** Mortality graphs of participants and refusals derived from Cox regression analysis and adjusted by sex. From age 75, Schoenfeld residuals indicated deviation from the assumption of proportional hazards (p<0.001), showing that the higher mortality among refusals was limited to the first 2 years, which is why a graph was not plotted. Mortality of 0–2 years, HR: 4.8, 95% CI 2.7 to 8.6; mortality of 3–5 years, HR: 1.3, 95% CI 0.9 to 1.9. (A) 2000: 5-year mortality from age 70, HR: 3.7, 95% CI 2.1 to 6.3, Schoenfeld residuals p=0.94. (B) 2009: 5-year mortality from age 79, HR: 1.6, 95% CI 1.2 to 2.1, Schoenfeld residuals p=0.29. (C) 2015: 5-year mortality from age 85, HR: 1.6, 95% CI 1.2 to 2.1, Schoenfeld residuals p=1.00. (D) 2018: 2-year mortality from age 88, HR: 2.8, 95% CI 1.7 to 4.5, Schoenfeld residuals p=0.60. HRs are for refusals compared with participants.

**Table 2** Characteristics of participants and refusals and same-aged individuals in Gothenburg and Sweden by age and examination year

| | 70-year-olds in 2000 | | | | 75-year-olds in 2005 | | | | 79-year-olds in 2009 | | | |
|---|---|---|---|---|---|---|---|---|---|---|---|---|
| | Participants | Refusals | Gothenburg | Sweden | Participants | Refusals | Gothenburg | Sweden | Participants | Refusals | Gothenburg | Sweden |
| n | 524† | 229 | 3508 | 75 468 | 767† | 429 | 3075 | 66 855 | 580 | 370 | 2647 | 58 184 |
| **Sociodemographic factors** | | | | | | | | | | | | |
| **Marital status, % (n)** | | | | | | | | | | | | |
| Married | 59.5 (311) | 53.3 (122) | 56.1 (1968) | 61.9 (46 697) | 54.6 (418) | 47.8 (205)* | 52.4 (1611) | 55.7 (37 238) | 49.5 (287) | 44.3 (164) | 46.8 (1238) | 49.4 (28 736) |
| Widowed | 14.7 (77) | 14.8 (34) | 15.8 (553) | 16.3 (12 291) | 22.4 (171) | 21.7 (93) | 21.5 (661) | 23.5 (15 715) | 28.6 (166) | 27.8 (103) | 28.3 (748) | 30.9 (17 973) |
| Divorced | 19.3 (101) | 17.0 (39) | 19.4 (679) | 14.0 (10 576)*** | 17.1 (131) | 21.7 (93) | 18.2 (560) | 13.5 (9051)** | 15.3 (89) | 20.3 (75) | 17.9 (473) | 13.0 (7550) |
| Never married | 6.5 (34) | 14.8 (34)*** | 8.8 (308) | 7.8 (5904) | 5.9 (45) | 8.9 (38) | 7.9 (243) | 7.3 (4851) | 6.6 (38) | 7.6 (28) | 7.1 (188) | 6.7 (3925) |
| **Education, % (n)** | | | | | | | | | | | | |
| Elementary school | 48.3 (253) | 58.1 (133)* | 47.8 (1678) | 53.9 (40 694)** | 44.4 (340) | 58.7 (252)*** | 47.7 (1467) | 53.2 (35 584)*** | 40.3 (234) | 58.6 (217)*** | 45.6 (1206)* | 52.5 (30 556)*** |
| Upper secondary school | 33.8 (177) | 27.5 (63) | 32.8 (1149) | 30.5 (23 027) | 36.3 (278) | 24.7 (106)** | 32.7 (1005) | 30.7 (20 553)*** | 38.8 (225) | 26.8 (99)*** | 33.4 (883)* | 30.9 (17 986)*** |
| Higher education | 16.4 (86) | 11.4 (26) | 15.5 (545) | 13.6 (10 291) | 18.4 (141) | 12.8 (55)* | 16.0 (493) | 14.1 (9408)*** | 20.5 (119) | 13.2 (49)** | 17.5 (462) | 14.6 (8475)*** |
| Born in Sweden, % (n) | 83.8 (438) | 76.4 (175)* | 80.4 (2821) | 87.6 (66 124)*** | 82.7 (633) | 80.2 (344) | 80.5 (2474) | 87.7 (58 640)*** | 84.3 (489) | 86.2 (319) | 80.5 (2130)* | 87.7 (51 046)* |
| Income, mean (SD) | 11 469 (8677) | 10 214 (7958) | 10 253 (9401)** | 10 481 (30 318)* | 11 710 (8118) | 11 131 (7780) | 11 549 (8803) | 11 325 (17 293) | 14 420 (12 394) | 12 129 (7815)*** | 13 541 (12 292) | 12 756 (15 869)** |
| Paid labour, % (n) | 5.2 (27) | 4.4 (10) | 3.9 (138) | 3.8 (2850) | 2.5 (19) | 1.6 (7) | 2.4 (75) | 2.1 (1425) | 1.4 (8) | 1.0 (5) | 0.7 (19‡) | 1.0 (592) |
| **Hospital discharge diagnoses, % (n)** | | | | | | | | | | | | |
| Cancer | 7.4 (39) | 9.6 (22) | 9.0 (314) | 7.3 (5488) | 13.2 (101) | 12.4 (53) | 12.9 (397) | 10.6 (7088)* | 15.7 (91) | 13.5 (50) | 15.4 (408) | 13.5 (7851) |
| Neuropsychiatric diseases | 6.1 (32) | 14.4 (33)*** | 10.1 (2353)** | 8.4 (6357) | 11.6 (89) | 16.1 (69)* | 14.1 (432) | 12.1 (8070) | 16.4 (95) | 15.4 (57) | 18.3 (484) | 16.5 (9590) |
| Dementia | 0.8 (4) | 3.1 (7)* | 0.6 (20) | 0.5 (357) | 1.4 (11) | 1.9 (8) | 1.5 (45) | 0.9 (604) | 1.9 (11) | 0.8 (3) | 1.9 (50) | 1.8 (1019) |
| Cardiovascular diseases | 23.7 (124) | 34.9 (80)** | 26.4 (927) | 26.7 (20 109) | 36.9 (283) | 42.9 (184)* | 39.4 (1211) | 38.3 (25 618) | 47.2 (274) | 43.5 (161) | 49.5 (1310) | 49.9 (29 035) |
| Diabetes mellitus | 4.4 (23) | 4.8 (11) | 4.9 (171) | 4.9 (3685) | 8.7 (67) | 9.6 (41) | 8.1 (250) | 7.3 (4872) | 9.0 (52) | 7.3 (27) | 9.6 (254) | 9.8 (5695) |
| Unipolar depression | 0.8 (4) | 0.0 (0) | 1.5 (54) | 1.4 (1059) | 1.6 (12) | 2.8 (12)** | 2.7 (82) | 2.1 (1372) | 3.4 (20) | 3.5 (13) | 4.0 (106) | 2.8 (1620) |
| Alcohol-related disorders | 1.5 (8) | 5.2 (12)** | 1.9 (65) | 1.4 (1056) | 1.3 (10) | 1.9 (8) | 1.7 (51) | 1.4 (906) | 0.9 (5) | 2.7 (10)* | 1.7 (45) | 1.3 (743) |
| Ischaemic heart diseases | 10.3 (54) | 12.7 (29) | 11.5 (403) | 11.5 (8641) | 15.1 (116) | 16.3 (70) | 16.7 (512) | 16.3 (1087) | 17.2 (100) | 17.0 (63) | 19.1 (506) | 19.9 (11 589) |
| Cerebrovascular diseases | 2.7 (14) | 8.3 (19)*** | 5.4 (191)** | 5.9 (4422)** | 8.2 (63) | 8.4 (36) | 8.6 (264) | 9.1 (6054) | 10.7 (62) | 8.6 (32) | 11.9 (315) | 12.0 (6973) |
| Chronic obstructive pulmonary disease | 2.3 (12) | 7.4 (17)*** | 3.3 (115) | 2.2 (1619) | 4.2 (32) | 6.5 (28) | 4.6 (140) | 3.4 (2252) | 4.8 (28) | 4.3 (16) | 5.7 (151) | 4.6 (2682) |
| Osteoarthritis | 6.3 (33) | 2.2 (5)* | 4.7 (165) | 5.8 (4384) | 9.5 (73) | 5.1 (22)** | 7.8 (241) | 9.7 (6477) | 14.7 (85) | 7.0 (26)*** | 12.3 (325) | 13.8 (8029) |

| | 85-year-olds in 2015 | | | | 88-year-olds in 2018 | | | |
|---|---|---|---|---|---|---|---|---|
| | Participants | Refusals | Gothenburg | Sweden | Participants | Refusals | Gothenburg | Sweden |
| n | 416† | 256§ | 1808 | 40 316 | 258 | 247† | 1321 | 29 472 |
| **Sociodemographic factors** | | | | | | | | |
| **Marital status, % (n)** | | | | | | | | |
| Married | 37.8 (157) | 31.5 (80) | 35.4 (640) | 36.8 (14 824) | 26.7 (69) | 29.7 (73) | 30.0 (396) | 29.6 (8728) |
| Widowed | 42.4 (176) | 43.7 (111) | 42.3 (764) | 45.1 (18 172) | 56.2 (145) | 48.4 (119) | 49.3 (651)* | 53.3 (15 700) |
| Divorced | 14.5 (60) | 18.1 (46) | 16.8 (303) | 12.3 (4972) | 12.8 (33) | 16.3 (40) | 15.6 (206) | 11.8 (3463) |

Continued

**Table 2** Continued

| | 85-year-olds in 2015 | | | | 88-year-olds in 2018 | | | |
|---|---|---|---|---|---|---|---|---|
| | Participants | Refusals | Gothenburg | Sweden | Participants | Refusals | Gothenburg | Sweden |
| n | 416† | 256§ | 1808 | 40 316 | 258 | 247† | 1321 | 29 472 |
| Never married | 5.3 (22) | 6.7 (17) | 5.6 (101) | 5.8 (2348) | 4.3 (11) | 5.7 (14) | 5.1 (68) | 5.4 (1581) |
| Education, % (n) | | | | | | | | |
| Elementary school | 37.8 (157) | 55.1 (140)*** | 43.6 (788)* | 50.8 (20 499)*** | 37.6 (97) | 48.4 (119)* | 42.2 (558) | 49.7 (14 642)*** |
| Upper secondary school | 40.0 (166) | 29.9 (76)** | 34.1 (617)* | 31.5 (12 687)*** | 36.4 (94) | 33.3 (82) | 34.5 (456) | 31.7 (9347) |
| Higher education | 20.7 (86) | 13.0 (33)* | 18.5 (335) | 15.7 (6323)** | 23.6 (61) | 14.6 (36)* | 19.1 (252) | 16.6 (4879)** |
| Born in Sweden, % (n) | 85.3 (354) | 83.5 (212) | 80.9 (1463)* | 87.9 (35 445) | 83.3 (215) | 82.5 (203) | 80.5 (1064) | 88.0 (25 930)* |
| Income, mean (SD) | 15 860 (9759) | 15 116 (20 451) | 15 827 (22 202) | 15 111 (21 828) | 19 315 (25 829) | 17 590 (20 007) | 17 312 (20 615) | 16 841 (25 586) |
| Paid labour, % (n) | N/A | N/A | N/A | N/A | N/A | N/A | N/A | N/A |
| Hospital discharge diagnoses, % (n) | | | | | | | | |
| Cancer | 20.2 (84) | 18.0 (46) | 20.1 (364) | 17.9 (7236) | 19.8 (51) | 20.2 (50) | 20.9 (276) | 19.1 (5634) |
| Neuropsychiatric diseases | 23.3 (97) | 22.3 (57) | 24.7 (446) | 25.4 (10 245) | 24.8 (64) | 34.8 (86)* | 27.7 (366) | 29.0 (8543) |
| Dementia | 3.1 (13) | 1.6 (4) | 2.0 (36) | 3.5 (1413) | <4¶ | 4.0 (10) | 2.0 (27) | 4.4 (1289) |
| Cardiovascular diseases | 62.3 (259) | 63.7 (163) | 63.6 (1150) | 66.1 (26 660) | 65.1 (168) | 70.4 (174) | 68.8 (909) | 72.2 (21 277)* |
| Diabetes mellitus | 10.6 (44) | 9.4 (24) | 11.1 (200) | 12.8 (5176) | 12.0 (31) | 12.1 (30) | 11.3 (149) | 13.6 (4019) |
| Unipolar depression | 4.6 (19) | 4.7 (12) | 5.3 (95) | 3.9 (1555) | 5.8 (15) | 5.3 (13) | 5.5 (73) | 4.0 (1191) |
| Alcohol-related disorders | <4¶ | <4¶ | 1.2 (21) | 1.0 (397) | 0.0 (0) | <4¶ | 1.1 (14) | 0.8 (242) |
| Ischaemic heart diseases | 18.3 (76) | 9.9 (51) | 21.8 (395) | 24.1 (9698)** | 17.8 (46) | 20.6 (51) | 21.3 (282) | 25.1 (7405)** |
| Cerebrovascular diseases | 12.7 (53) | 17.2 (44) | 15.0 (272) | 16.1 (6503) | 14.7 (38) | 16.2 (40) | 15.8 (209) | 17.1 (5044) |
| Chronic obstructive pulmonary disease | 4.8 (20) | 4.7 (12) | 5.6 (101) | 6.0 (2405) | 5.4 (14) | 2.8 (7) | 4.9 (65) | 6.1 (1793) |
| Osteoarthritis | 20.0 (83) | 13.3 (34)* | 18.5 (335) | 19.5 (7854) | 24.0 (62) | 15.4 (38)* | 20.9 (276) | 21.3 (6268) |

Comparisons made between participants and refusals, Gothenburg and Sweden.
Statistical analysis: to test differences between groups, t-test or $\chi^2$ test was used.
Values in bold are significant at p<0.05; all other values are not significant.
Values in italics are not exact (true value within a 1–3 range) due to protection of anonymity.
N/A means not applicable as the cut-off age for paid labour in Statistics Sweden registers is 84.
*P<0.05, **P<0.01, ***P<0.001.
†One individual is missing in the sociodemographic variables because the personal identity number was not found in the register.
‡Information is missing on three individuals (all men).
§Two individuals are missing in the sociodemographic variables because their personal identity number was not found in the register.
¶Less than four individuals. Exact number not presented to protect anonymity. Statistical comparison not performed.

with the same-aged population for the entire country of Sweden.

This study has both strengths and limitations. The strengths of this study include the five timepoints of cross-sectional health examinations and the use of register data to compare participants not only with refusals, but also with same-aged individuals from Gothenburg and same-aged individuals from the entire country. There are also several limitations. First, the same individuals are included at repeated examinations and might thus be influenced by the Hawthorne effect, that is, they might change behaviours as a result of their participation, resulting in increased differences between participants and refusals.[11] Second, participants make up 15%–25% of the total comparison group of individuals in Gothenburg. This made the comparison group more similar to participants. On the other hand, if we excluded participants, the comparison group would have been less representative of same-aged individuals in Gothenburg. Third, using register data to compare participants with the target population might miss important differences. A previous study found only minor differences in disease burden between participants and non-participants when using registers, but large differences when using clinical and biochemical variables from a baseline examination conducted 20 years earlier.[12] It should also be noted that the sensitivity of disorders in the NPR varies, with higher sensitivity for more severe disorders and less for milder disorders.[9 13] Fourth, in some years, the examinations were delayed due to logistic issues, creating a higher mean age in the population examined than was targeted. This increased the number of individuals who died or developed disorders before the examination, a bias which becomes more important with age. Finally, a small group was excluded due to either not speaking Swedish (1.6%–4.1%) or not being traceable (0.5%–3.6%). These groups were, however, small and the exclusion probably did not affect the main results.

The participation rate was initially 70% at age 70, declined to 61%–64% at ages 75–85, and dropped to 51% at age 88. Response rates between 50% and 70% have previously been reported in similar Swedish population-based studies of adults aged 65 years and older,[14] with indications that non-response was higher among the oldest population. Possible reasons for declining response rate with advancing age may be participation exhaustion, an age effect or a period effect. Participation exhaustion might occur due to a high number of extensive examinations. To minimise this effect, we offered to perform the examination at the participant's homes or to split up the examinations at different occasions.[15] However, the response rates in our previous studies rather support an age effect where the response rate among the 70-year-olds born in 1944 and 1930 was around 70%[4] and among the 85-year-olds born in 1923–1924 and 1930 around 60%.[6] Other explanations for an age effect, also reported by others,[16–18] include cognitive impairment,[17 18] frailty and illness,[17] which all increase with age. Population-based

studies in older populations might thus be less representative with increasing age. Support for a period effect is suggested by the declining response rates in research studies over the last decades.[1] For example, the first H70 examination conducted in 1971 on 70-year-olds had a response rate of 85%.[5] At that examination, participants and non-participants were very similar, in contrast to the examination of 70-year-olds in our study. Another example is a Finnish health survey[19] conducted in 1987–2012. They reported a declining participation rate over time in all socioeconomic groups, but a steeper decline among those with low educational level. This resulted in increased differences between participants and non-participants.[19] The reasons for period effects include an overall decrease in social participation, increased complexity of modern life, lower trust in society, science and research, increased burden due to increasing number of invitations to participate in studies, and increasingly demanding research projects.[7] This increases the risk of healthy responder bias, making the results from population-based studies less generalisable and cohort comparisons more difficult to perform.

Our findings that participants in general were healthier and had higher education and income than refusals are in line with a number of population-based studies. For example, participants compared with refusals are reported to be younger,[16–18] more often cohabiting or married,[16 20–22] born in the study country or in neighbouring countries,[20 21] have higher income,[21] educational level[20 22] and socioeconomic status,[12] have better health status,[12 23] less often have a diagnosis in hospital discharge registers,[12 16 20] and have lower mortality rates.[12 16] Among older persons, participants reportedly have less cognitive impairment,[17 18] illness and frailty.[17] Interestingly, refusals less often had osteoarthritis in the NPR compared with participants at all examinations, in contrast to what was observed for most other disorders. Osteoarthritis is a disorder with a fairly large proportion of undiagnosed cases.[24] It is thus possible that those not seeking medical attention or receiving specialised care for osteoarthritis are also less prone to participate in health examination studies.

We also examined differences between participants and the total population of same-aged individuals in Gothenburg, that is, the target population, to get a better appreciation of external validity than is achieved by only examining differences compared with refusals. Our finding that participants were more similar to the target population than to refusals, especially at younger ages, is in line with previous studies.[12 22] However, there were some differences. Participants had higher educational level and were more often born in Sweden at ages 79 and 85, less often had cerebrovascular and neuropsychiatric disorders and higher mean income at age 70, and were more often widowed at age 88. The proportion being excluded due to not speaking Swedish was larger at ages 79 and 85 compared with the other examination years. This might explain why participants compared with

same-aged individuals in Gothenburg were more often born in Sweden at these ages.

There were more differences between participants and same-aged individuals in Sweden. This is not surprising as Gothenburg is one of Sweden's largest cities and is a university city. In addition, it has a higher proportion of individuals born outside Sweden and higher educational level than the rural areas of Sweden. This illustrates the difficulty of generalising findings from a local target population to even a relatively small and homogenous population, as Sweden.

We found no significant differences in response rates between women and men, in contrast to other population-based studies, which report higher response rates among women.[16 20 22 25] However, in two of these studies, there were no sex differences at higher ages,[16 25] and the other two only included individuals aged below 64 and 54 years, respectively.[20 22] Some studies report that sex differences in response rates diminish with age.[22 26]

Our study thus shows that, although there are differences between participants and refusals, participants are more similar to the target population overall. To further minimise potential selection bias in future studies on older adults, adjusting the examinations to meet the needs of those more prone to decline might be an alternative. In the present study, home visits were offered to those who did not want to visit the clinic.[15] However, further adjustments such as shorter examinations, telephone interviews or questionnaires to those who are hesitant of participation might be applied. This would, however, potentially induce measurement errors. There are also statistical methods to potentially account for missing data, such as multiple imputation or weighting methods, which could counteract some of the issues of selection bias.[27]

## CONCLUSION

In conclusion, our data show that participants were similar to the target population of Gothenburg. However, participants differed most compared with refusals, and to a certain extent to same-aged individuals in Sweden, indicating that the studies might underestimate the burden of disease. These results contribute unique and up-to-date knowledge on participation bias in these high age groups and show the importance of having different comparison groups when assessing representativeness. This finding is readily transferable to similar studies, and since population-based studies on older adults contribute information on disease distribution and knowledge of risk–disease associations it is important to be aware of potential limitations in the representativeness that might hamper generalisability.

## Author affiliations
[1]Department of Psychiatry and Neurochemistry, University of Gothenburg, Institute of Neuroscience and Physiology, Gothenburg, Sweden
[2]Centre for Ageing and Health (AGECAP), University of Gothenburg, Gothenburg, Sweden
[3]Department of Mood disorders, Sahlgrenska University Hospital, Region Västra Götaland, Gothenburg, Sweden
[4]Psychiatry, Cognition and Old Age Psychiatry Clinic, Sahlgrenska University Hospital, Region Västra Götaland, Gothenburg, Sweden
[5]Aging Research Center, Department of Neurobiology, Care Sciences and Society, Karolinska Institutet and Stockholm University, Stockholm, Sweden
[6]Department of Psychiatry, Psychotic Disorders, Sahlgrenska University Hospital, Region Västra Götaland, Mölndal, Sweden

**Acknowledgements** The authors would like to thank all participants of the H70 studies in Gothenburg and the research group members for their cooperation in data collection and management. The authors also acknowledge seminal contributions to this study by the late Dr Svante Östling.

**Contributors** HW, LR and IS had full access to all the data in the study and take responsibility for the integrity of the data and the accuracy of the data analysis. Concept and design: HW, LR and IS. Acquisition, analysis or interpretation of data: HW, LR, FA, HF, PG, XG, EJ, LJ, SK, MF, JN, MR, TRS, JSa, SS, RS, JSk, MW, AZ and IS. Drafting of the manuscript: HW, LR and IS. Statistical analysis: HW and LR. Principal investigator: IS. Critical revision of the manuscript for important intellectual content: HW, LR, FA, HF, PG, XG, EJ, LJ, SK, MF, JN, MR, TRS, JSa, SS, RS, JSk, MW, AZ and IS. HW is the guarantor.

**Funding** The study was financed by grants from the Swedish state under the agreement between the Swedish government and the county councils, the ALF Agreement (ALF 716681); Stena Foundation; Swedish Research Council (11267, 2005-8460, 2007-7462, 2012-5041, 2015-02830, 2019-01096, 2013-8717, NEAR 2017-00639); Swedish Research Council for Health, Working Life and Welfare (2004-0145, 2006-0596, 2008-1111, 2010-0870, 2013-1202, 2018-00471, 2001-2646, 2003-0234, 2004-0150, 2006-0020, 2008-1229, 2012-1138, AGECAP 2013-2300, 2013-2496); Konung Gustaf V:s och Drottning Victorias Frimurarestiftelse; Hjärnfonden (FO2014-0207, FO2016-0214, FO2018-0214, FO2019-0163, FO2020-0235); Alzheimerfonden (AF-554461, AF-647651, AF-743701, AF-844671, AF-930868, AF-940139); Eivind och Elsa K:son Sylvans stiftelse; the Alzheimer's Association Zenith Award (ZEN-01-3151); the Alzheimer's Association Stephanie B Overstreet Scholars (IIRG-00-2159); the Bank of Sweden Tercentenary Foundation; Stiftelsen Söderström-Königska Sjukhemmet; Stiftelsen för Gamla Tjänarinnor; and Handlanden Hjalmar Svenssons Forskningsfond. SK was financed by grants from the Swedish state under the agreement between the Swedish government and the county councils, the ALF Agreement (ALFGBG-81392, ALF GBG-771071); Alzheimerfonden (AF-842471, AF-737641); the Swedish Research Council (2019-02075); Stiftelsen Demensfonden; Stiftelsen Hjalmar Svenssons Forskningsfond; and Stiftelsen Wilhelm och Martina Lundgrens vetenskapsfond. SS was financed by grants from the ALF Agreement (ALFGBG-637271) and the Sahlgrenska Academy Homecoming Fellowship (Dnr V2012/294).

**Competing interests** None declared.

**Patient and public involvement** Patients and/or the public were not involved in the design, or conduct, or reporting, or dissemination plans of this research.

**Patient consent for publication** Not required.

**Ethics approval** This study involves human participants and the Regional Ethical Review Board approved the H70 studies in Gothenburg (approval numbers: 240800/S227-00, 041104/T453-04, 090309/075-09, 270415/131-15 and 230418/278-18). The Swedish Ethical Review Authority approved the publication of aggregated data on the individuals who declined participation in the study (approval numbers: 170620/2020-02181 and 170620/2020-02188). Participants gave informed consent to participate in the study before taking part.

**Provenance and peer review** Not commissioned; externally peer reviewed.

**Data availability statement** Data are available upon reasonable request.

**ORCID iDs**
Hanna Wetterberg http://orcid.org/0000-0003-3836-825X
Lina Rydén http://orcid.org/0000-0002-3817-2620
Felicia Ahlner http://orcid.org/0000-0003-3116-7332
Hanna Falk Erhag http://orcid.org/0000-0002-6494-2895
Pia Gudmundsson http://orcid.org/0000-0001-7173-4341
Xinxin Guo http://orcid.org/0000-0002-1353-287X
Erik Joas http://orcid.org/0000-0001-9546-2192
Lena Johansson http://orcid.org/0000-0002-9928-4909
Silke Kern http://orcid.org/0000-0001-7617-7214
Madeleine Mellqvist Fässberg http://orcid.org/0000-0002-7366-4444
Jenna Najar http://orcid.org/0000-0002-1762-4413
Mats Ribbe http://orcid.org/0000-0002-5998-656X
Therese Rydberg Sterner http://orcid.org/0000-0001-8202-8522
Jessica Samuelsson http://orcid.org/0000-0002-8891-6720
Simona Sacuiu http://orcid.org/0000-0003-0472-7699
Robert Sigström http://orcid.org/0000-0003-1505-1532
Johan Skoog http://orcid.org/0000-0001-9549-5875
Margda Waern http://orcid.org/0000-0002-8330-6915
Anna Zettergren http://orcid.org/0000-0002-7182-8417
Ingmar Skoog http://orcid.org/0000-0002-9803-3438

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
