## [Reviewer comments · BMJ Open]

ARTICLE DETAILS

TITLE (PROVISIONAL)	Representativeness in population-based studies of older adults – Five waves of cross-sectional examinations in the Gothenburg H70 Birth Cohort Study
AUTHORS	Wetterberg, Hanna; Rydén, Lina; Ahlner, Felicia; Falk, Hanna; Gudmundsson, Pia; Guo, Xinxin; Joas, Erik; Johansson, Lena; Kern, Silke; Fässberg, Madeleine; Najjar, Jenna; Ribbe, Mats; Sterner, Therese Rydberg; Samuelsson, Jessica; Sacuiu, S; Sigström, Robert; Skoog, Johan; Waern, M; Zettergren, Anna; Skoog, Ingmar

VERSION 1 – REVIEW

REVIEWER	Lee, Yunhwan Ajou University School of Medicine and Graduate School of Medicine, Preventive Medicine and Public Health
REVIEW RETURNED	03-Oct-2022

GENERAL COMMENTS	This study assessed the representativeness of the H70 study cohorts by comparing the participation rate of the study respondents with that of refusals and same-aged individuals in Gothenburg and Sweden. Significant differences were found due to the age effect and population characteristics, implying close attention in generalizing the study results. The paper provides a detailed account of the selection bias in this population-based study, in general agreement with previous studies. I have several questions or points for the authors to consider. p. 10, ln. 32: “There were no differences in response rates between women and men ...” In 2000, however, the response rate was higher in women (72.1%) than in men (66.9%). Slightly more men (63.2%) than women (59.5%) responded in 2009. Wasn’t this gender difference significant? Please explain. (cf. p. 21, ln. 16: “We found no differences in response rates between women and men...”) p. 19, ln. 9: Was there any seasonal effects in the differences in the participation rate? As the authors mentioned, some surveys were delayed due to logistics. I would like to see more discussions on the implications of the study. For example, what do the authors propose to take into account the selection bias in generalizing the results?
---

REVIEWER	Yeo, Anthony Western Sydney University
REVIEW RETURNED	11-Oct-2022

GENERAL COMMENTS	You mentioned that you did a Cox's proportional hazard adjusted for sex. I would like to see this graph. In this graph, you should have 2 curves, one for men, one for women and you will compare the both of them, as you stated that you did and provide the Schoenfeld residuals/results to verify that the assumption of proportional hazards have not been violated. You had mentioned this in your methods section but I did not see KM curves in the paper. Optional, if you like you can also include a curve with both men and women combined.
---

VERSION 1 – AUTHOR RESPONSE

Reviewer: 1

Dr. Yunhwan Lee, Ajou University School of Medicine and Graduate School of Medicine Comments to the Author:

This study assessed the representativeness of the H70 study cohorts by comparing the participation rate of the study respondents with that of refusals and same-aged individuals in Gothenburg and Sweden. Significant differences were found due to the age effect and population characteristics, implying close attention in generalizing the study results. The paper provides a detailed account of the selection bias in this population-based study, in general agreement with previous studies.

I have several questions or points for the authors to consider.

1) p. 10, ln. 32: “There were no differences in response rates between women and men ...” In 2000, however, the response rate was higher in women (72.1%) than in men (66.9%). Slightly more men (63.2%) than women (59.5%) responded in 2009. Wasn't this gender difference significant? Please explain. (cf. p. 21, ln. 16: “We found no differences in response rates between women and men...”)

Response: Thank you for highlighting this difference. The slightly differing response rates were not significantly different at any examination year. We have clarified this in the results (“There were no **significant** differences in response rates between women and men, although women had slightly higher response rates in 2000 (72.1% versus 66.9%, $p=0.122$) and lower in 2009 (59.5% versus 63.2%, $p=0.254$) (see Additional file 2, Figure s1”). In the discussion, we clarified this by adding “significant”: “We found no **significant** differences in response rates between women and men,...”

2) p. 19, ln. 9: Was there any seasonal effects in the differences in the participation rate? As the authors mentioned, some surveys were delayed due to logistics.

Response: The effect of season on participation rate is indeed very interesting. We tested this by combining data from the examinations in 2009, 2015 and 2018 (data not available at examinations 2000 and 2005) and compared the response rates by season with Chi-square test. When defining the season of first contact attempt in weeks 6-22 as spring, weeks 22-35 as summer, weeks 36-47 as autumn and weeks 48-5 as winter, we did not see a difference in response rate ($X^2(3, N = 1807) = 2.56, p = 0.464$).

3) I would like to see more discussions on the implications of the study. For example, what do the authors propose to take into account the selection bias in generalizing the results?

Response: Thank you for this comment. We agree that implications of the study needs to be further discussed in the paper. We have added the following section in the end of the discussion:

Our study thus shows that although there are differences between participants and refusals, participants are more similar to the target population overall. To further minimize potential selection bias in future studies on older adults, adjusting the examinations to meet the needs of those more prone to decline might be an alternative. In the present study, home visits were offered to those who did not want to visit the clinic (15). However, further adjustments, such as shorter examinations, telephone interviews or questionnaires to those who are hesitant of

participation might be applied. This would, however, potentially induce measurement errors. There are also statistical methods to potentially account for missing data, such as multiple imputation or weighting methods that could counteract some of the issues of selection bias (27).

Reviewer: 2

Dr. Anthony Yeo, Western Sydney University Comments to the Author:

Dear corresponding author:

1) You mentioned that you did a Cox's proportional hazard adjusted for sex. I would like to see this graph. In this graph, you should have 2 curves, one for men, one for women and you will compare the both of them, as you stated that you did and provide the Schoenfeld residuals/results to verify that the assumption of proportional hazards have not been violated. You had mentioned this in your methods section but I did not see KM curves in the paper. Optional, if you like you can also include a curve with both men and women combined.

Response: Yes, we performed Cox's proportional hazard models to compare 5-year (and 2-year for the 88-year olds) mortality between participants and refusals from all examination points, i.e. five models. We have exchanged Table 2 containing only the hazards ratios in favour of the figures below, illustrating the hazards. The hazard ratios and the Schoenfeld residuals can be found in the figure legend.

Figure legend

Mortality graphs in participants and refusals, derived from Cox regression analysis, adjusted by sex.

HR: Hazard ratio refusals compared to participants, CI: confidence interval.

From age 75, Schoenfeld residuals indicated deviation from the assumption of proportional hazards ($p < 0.001$), showing that the higher mortality among refusals was limited to the first 2 years, which is why a graph was not plotted. The 0-2 year mortality HR: 4.8, 95% CI: 2.7-8.6, and 3-5 year mortality HR: 1.3, 95% CI: 0.9-1.9.

a) 2000: 5-year mortality from age 70, HR: 3.7, 95% CI: 2.1-6.3, Schoenfeld residuals $p = 0.94$.

b) 2009: 5-year mortality from age 79, HR: 1.6, 95% CI: 1.2-2.1, Schoenfeld residuals $p = 0.29$.

c) 2015: 5-year mortality from age 85, HR: 1.6, 95% CI: 1.2-2.1, Schoenfeld residuals $p = 1.00$.

d) 2018: 2-year mortality from age 88, HR: 2.8, 95% CI: 1.7-4.5, Schoenfeld residuals $p = 0.60$.

VERSION 2 – REVIEW

REVIEWER	Lee, Yunhwan Ajou University School of Medicine and Graduate School of Medicine, Preventive Medicine and Public Health
REVIEW RETURNED	25-Nov-2022

GENERAL COMMENTS	The authors have sufficiently addressed all my concerns.
--

REVIEWER	Yeo, Anthony Western Sydney University
REVIEW RETURNED	14-Nov-2022

GENERAL COMMENTS

You have provided the KM graph with the appropriate curves and statistics. Thank you.